# Pollination Syndrome, Florivory, and Breeding System of *Satyrium nepalense* var. *ciliatum* (Orchidaceae) in Central Yunnan, China

**DOI:** 10.3390/plants13091228

**Published:** 2024-04-28

**Authors:** Lei Tao, Kaifeng Tao, Qingqing Li, Yingduo Zhang, Xiangke Hu, Yan Luo, Lu Li

**Affiliations:** 1College of Forestry, Southwest Forestry University, Kunming 650224, China; tl0427@126.com (L.T.); 17856109959@163.com (K.T.); ydzhang@genedenovo.com (Y.Z.); hxk1208@126.com (X.H.); 2College of Biological Science and Food Engineering, Southwest Forestry University, Kunming 650224, China; doublelqq@163.com; 3Kunming Xianghao Technology Co., Ltd., Kunming 650204, China; 4Southeast Asia Biodiversity Research Institute, Chinese Academy of Sciences & Center for Integrative Conservation, Xishuangbanna Tropical Botanical Garden, Chinese Academy of Sciences, Mengla 666303, China

**Keywords:** ant, apomixis, herbivory, floral anatomy, nectar, Orchidaceae, *Satyrium*

## Abstract

Research on *Satyrium nepalense* var. *ciliatum* (Lindl.) Hook. f. has primarily focused on populations in Northwestern Yunnan, with limited studies on pollination syndromes and insect behavior. In addition, it is geographically limited in its breeding system studies. Here, pollination syndromes, florivory, and breeding systems of *S. nepalense* var. *ciliatum* from Liangwang Mountain (Central Yunnan, China) were investigated through field work, microscope, scanning electron microscope (SEM), and parafin section. It was revealed that the pollination syndrome was possessing out-crossing, such as bright color, a developed rostellum, nectar glands in the spur, and food hairs at the lip base. The color and nectar attracted flower visitors, and florivory was observed. Some flower visitors pollinated their companion species. Ants were identified as floral visitors for the first time in *Satyrium*, although substantial pollination was not observed. Ants might be potential pollinators. *S. nepalense* var. *ciliatum* possessed a mixed breeding system, including selfing, out-crossing, and apomixis, with apomixis being predominant in nature. It is suggested that the pollination syndrome, florivory, and pollination competition would contribute to its mixed breeding systems, particularly leading to the occurrence of apomixis.

## 1. Introduction

The genus *Satyrium* Sw. encompasses approximately 90 species, primarily found in Africa, with limited representation in Asia, including one endemic to China [1]. It attracts a diverse array of floral visitors and pollinators, including birds, bees, beetles, and moths, with associations linked to their respective pollination syndromes [2,3,4]. *S. nepalense* var. *ciliatum* is distributed in the southwestern regions of Guizhou, Northwestern parts of Hunan, Western and Southwestern areas of Sichuan, Eastern and Southeastern parts of Tibet, Northwestern and Southeastern regions of Yunnan, China, as well as in Bhutan, India, and Nepal [1]. It is the only species known to exhibit two distinct sexual forms: the female form and the hermaphroditic form [1,5]. One key feature distinguishing it from *S. nepalense* D. Don is the morphology of the floral spurs. In the former, the spurs are conic, stout, and shorter than the ovary, while in the latter, they are downcurved, cylindric, and slender, often as long as or longer than the ovary [1].

Furthermore, *Satyrium* is utilized for traditional medicinal purposes; for example, ancient Greek healers utilized *Satyrium* tubers as aphrodisiacs [6]. In India, traditional healthcare establishments employ *S. nepalense* tubers to create energy tonics and alleviate various fevers [7]. Despite the high medicinal value of *S. nepalense* var. *ciliatum* in Nepal, local awareness is lacking, leading to its underutilization [8]. The population of *Satyrium* resources is rapidly declining due to fragmented distribution and its endangered status, driven by the demand for herbal medicine and significant environmental degradation [9].

Pollination syndromes refer to a set of floral traits associated with attracting and utilizing specific groups of animal pollinators [10]. Among these floral traits, color, reward, fragrance, and floral morphology are the most well-known components of pollination syndromes [11]. *Satyrium* exhibits considerable diversity with significant variation in flower color, size, and shape, likely as adaptations to different pollinators [4,12,13]. The notable traits of plants within the genus *Satyrium* include resupinate flowers, a helmet-shaped and twin-spurred lip [1,14]. Despite this morphological diversity, some species share striking similarities, making them difficult to distinguish [15]. This may be attributed to convergent adaptations to the same pollinators or close phylogenetic relationships [4,13,16]. Presently, research on *Satyrium* is predominantly conducted in South Africa, where pollinators include bees, long-proboscid flies, beetles, day-flying hawkmoths, noctuid moths, nocturnal hawkmoths, butterflies, and sunbirds (references in [4]).

Currently, five species of *Satyrium* have been documented as being pollinated by Nectariniidae [4,17,18,19,20]. These species typically exhibit brightly colored and odorless flowers, with large, flattened viscidia [17,19]. Furthermore, sunbirds occasionally visit some *Satyrium* orchids, although these orchids have smaller viscidia adapted for insect pollination, and then swiftly scrape off the pollen adhering to their bills [21]. Interestingly, the length of the floral spur in these *Satyrium* orchids is not correlated with the length of bird bills, but rather with the elongated tongues of the birds [17].

Studies have shown a strong positive correlation between the length of the floral spur and the species of insect pollinators in the genus *Satyrium* [4]. Currently, the species with the longest floral spurs in this genus are pollinated by moths [22], while those with the shortest spurs are pollinated by flies and beetles, with floral spur lengths ranging from 10 to 15 mm in species pollinated by bees, and plants pollinated by bees often exhibit floral markings serving as nectar guides [4]. Additionally, moth-pollinated *Satyrium* orchids typically have pale white or greenish flowers, long floral spurs, rostellum lateral lobes with plate-like viscidia, and emit an evening scent, among other pollination syndromes [4,23]. Moreover, flies are also attracted to these orchids, with *S. pumilum* Thunb. attracting flesh flies (Sarcophagidae) due to emitting odors resembling those attractive to flesh flies [24]. Long-proboscid flies are attracted too, in the cases of *S. macrophyllum* Lindl. and high-altitude *S. neglectum* Schltr., which exhibit traits such as pink or creamy-white, odorless, and long-spurred flowers [4,25]. Beetles are attracted to *Satyrium* due to the emission of fruity odors and the availability of exposed nectar, and they may also nibble the food hairs of lips to access nectar [3]. The pollination syndrome traits for butterfly-pollinated species include having pink flowers [4].

The pollination syndrome is related to the breeding system [26]. *Satyrium* exhibit self-compatibility [27], but they are prone to experiencing inbreeding depression, resulting in fewer viable offspring from self-pollination [27,28]. For instance, the flowers of *S. longicauda* Lindl. display plasticity, increasing attractiveness through larger displays when pollinators are scarce, or limiting mating costs through smaller displays when pollinators are abundant [29]. In the absence of pollinators in the environment, *S. rupestre* Schltr. ex Bolus has developed mechanisms for automatic self-pollination [30]. *S. nepalense* exhibits a high natural fruit set rate (>90%), indicating its ability for autonomous self-pollination [31].

Current studies on *Satyrium nepalense* var. *ciliatum* primarily focuses on reproductive biology, particularly in populations located in Northwestern and Southeastern Yunnan, China [32,33,34,35,36]. Based on flower morphology, these populations have been classified into six types [32]. Bagging experiments conducted on populations in Northwestern Yunnan suggest a mixed breeding system with high fruit set rates [32,33]. Additionally, hermaphroditic populations contribute more to genetic diversity than gynodioecious populations, with female flowers in all-female populations typically occurring in drier locations compared to hermaphroditic individuals [35,37]. These plants exhibit lower sexual dimorphism in nutrient traits compared to reproductive traits and have a higher level of phenotypic integration, indicating a negative correlation between sexual dimorphism and phenotypic integration levels [36]. However, there is limited research on pollination biology, with only one instance of bee pollination observed in Northwestern Yunnan populations over several years, along with instances of insect florivory, and female plants showing greater adaptability to being eaten [34,36].

Given that current research on *Satyrium nepalense* var. *ciliatum* is predominantly focused on populations in Northwestern Yunnan and mainly encompasses reproductive biology, there is a lack of studies on pollination syndromes and insect behavior. Additionally, research on its breeding system is limited by geographical constraints. The objectives of the study are as follows: (1) to analyze the pollination syndromes of *S. nepalense* var. *ciliatum*; (2) to explore the relationship between pollination syndromes and insect visitation behavior; and (3) to investigate the relationship between pollination syndromes, insect visitation behavior, and the formation of the breeding systems in populations in Central Yunnan. This research aims to provide new scientific evidence for the study of pollination biology and the formation of the breeding system in *S. nepalense* var. *ciliatum* populations in different regions.

## 2. Results

### 2.1. Pollination Syndrome

*Satyrium nepalense* var. *ciliatum* is a terrestrial orchid with an inflorescence consisting of 22 flowers on average. The flowers were pink and had reflexed and foliaceous bracts (Figure 1a). The lip was superior and deeply hooded, measuring an average length of 5.45 ± 0.21 mm and an average width of 4.61 ± 0.09 mm. Two spurs adnate to the lip base, with an average length of 5.19 ± 0.02 mm and an average diameter of 1.15 ± 0.04 mm. The dorsal sepal appeared linear-oblong, while the lateral sepals were oblong-spatulate. (Figure 1b,c). The lip, dorsal sepal, and petals constituted a flower channel, measuring 4.75 ± 0.10 mm in length and 2.56 ± 0.10 mm in width (Figure 1b,c). The column was elongated and incurved. The stigma presented a subquadrate shape, and the rostellum featured three lobes, forming an angle of 166.6° (Figure 1d,e). The rostellum was well developed, positioning the anther below it (Figure 1d,e). A flower has a pair of sectile pollinia in a ripped anther, each with a caudicle and a viscidium (Figure 1f). The sectile pollinia were composed of massulae, and their pollen exine sculpture exhibited reticulate (Figure 1g–i).

Humans cannot detect the smell of *Satyrium nepalense* var. *ciliatum*. In order to investigate whether the species possesses substances such as nectar to attract flower visitors, the flower structure was examined using SEM and histochemical staining. The results revealed that the abaxial epidermis of the bract possessed sugar-containing stomata, while sugar-containing food hairs were present on the bract margins (Figure 2a–f). These food hairs were also distributed at the base of the lip (Figure 2g–i). Furthermore, nectar deposition occurred at the base of the spur, with the inner wall of the spur containing nectar glands (Figure 2j–o). These nectar glands were of three types: villiform, dome-shaped, and obpyriform (Figure 2m–o). Secretory cells were found at the base of the nectar glands (Figure 2m,n). To investigate the variation in nectar volume over the course of a day, measurements were taken at four different time intervals. The findings revealed a dynamic pattern in the nectar volume of the species throughout the daytime. Specifically, there was an increase observed from 9:00 a.m. to 3:00 p.m., followed by a decrease from 3:00 p.m. to 5:00 p.m. The peak nectar content was recorded at 3:00 p.m., reaching 1.07 ± 0.08 μL on average, whereas the minimum nectar content was observed at 9:00 a.m., measuring 0.72 ± 0.13 μL on average.

### 2.2. Florivory and Flower Visitor Observation

Because *S. nepalense* var. *ciliatum* exhibited an out-crossing trend pollination syndrome and had a food reward, in order to investigate florivory in *S. nepalense* var. *ciliatum*, observations were made on florivorous plants and their specific structures. The results showed that 43 out of 50 observed plants displayed signs of being consumed. Florivory was evident from the bud stage to the mature flower stage, affecting various floral structures, including bracts, lips, petals, gynostemium, and even the ovary (Figure 3a–d). Upon observing flowers where the ovary remained unharmed despite florivory, all of these flowers exhibited ovary enlargement and fruit production. Interestingly, even when the gynostemium structures, especially pollen and the stigma, were subjected to florivory, the ovary continued to enlarge and produce fruits. In addition, sectile pollinia were observed exposed and adhered to the lip and petals (Figure 3i).

Due to the widespread occurrence of florivory, observations were conducted on the floral visitors of *Satyrium nepalense* var. *ciliatum*. The results showed that *S. nepalense* var. *ciliatum* attracted many visiting insects, with the most common being dipterans, including flies (Figure 3k,l). The frequency of their visits to the flowers gradually increased from 8:00 to 15:00, reaching a peak between 15:00 and 16:00, and then gradually decreased thereafter (Figure 4). These insects utilized their mouthparts to access different flowers parts, but they did not enter the flower. Observations included beetles visiting the flowers, which did not enter the flower but rather fed on the ovary (Figure 3j). Bees, wasps, and bumblebees were observed approaching the flowers, but they quickly flew away without entering the flowers or carrying pollen. Additionally, upon dissecting some flowers, it was discovered that worms were found in the florivorous flowers (Figure 3e,f), and Thripidae were observed in the ovary (Figure 3g,h). Also, a locust larva was seen visiting the flower (Figure 3m).

Additionally, companion species were found nearby the plants of *Satyrium nepalense* var. *ciliatum*, such as *Pedicularis salviiflora* Franch. ex F. B. Forbes & Hemsl, *Geranium refractum* Edgew. & Hook. f., and *Aster yunnanensis* Franch. And many insects were observed visiting these companion species, including bumblebees and flies (Figure 3n–p).

As ants were found entering the flowers, observations were conducted on the two species of ants and their flower-visiting behavior. Two ant species, *Pheidole zoceana* Santschi (Figure 4f) and *Temnothorax* sp. (Figure 4g), were identified and observed engaging in floral visitation, exhibiting similar visiting behaviors (Figure 4). The frequency of their visits to the flowers increased from 8:00 to 12:00, then decreased until 14:00, and then increased again until 15:00. Between 15:00 and 16:00, it reached a peak, and then gradually decreased thereafter (Figure 5). The ants climbed up the flower stems and entered the flowers along the petals, then they reached the spur to feed on the nectar. They did not choose a specific side to enter the flowers. As they moved inside the flowers, they could easily switch from one side of the spur to the other due to gaps between the central column and the lip (Figure 4d,e). Additionally, while crawling inside the flowers, ants passed over the stigma and rostellum (Figure 4c). After finishing their feeding, the ants exited the flowers and moved on to the next ones for more foraging. It was worth noting that as long as there was nectar inside the spur, even in florivorous flowers, the ants still visited them for feeding (Figure 4a,b). Another interesting point was that the ants’ flower-visiting behavior was not disturbed by other floral visitors (Figure 4a).

As ants were observed entering the flowers, they were captured to detect if they carried pollen. The results indicated that some ants carried pollens, although in small numbers (*P. zoceana:* 3/10, *Temnothorax* sp.: 2/7), and they only carried massulae rather than entire pollinia (Figure 4f,g). However, substantial evidence of ants carrying pollens to the stigma and completing the pollination process was not observed during field observations.

### 2.3. Breeding System

The observation in the field revealed that some flowers had florivorous stigma and pollinia that were chewed on by animals, yet the ovaries still developed into fruits. Subsequently, bagging experiments were conducted to observe their reproductive systems. The results showed that the fruit set in autonomous self-pollination was found to be 94.33% (50/53), manual self-pollination resulted in 93.75% (45/48) fruit set, geitonogamy resulted in 94.82% (55/58) fruit set, out-crossing resulted in 94.87% (37/39) fruit set, apomixis showed a fruit set of 95.71% (67/70), and the natural control had a fruit set of 95.33% (143/150). In its natural state, the observed pollen removal rate for *Satyrium nepalense* var. *ciliatum* was 25% (50/200). All the fruit set of artificial treatments did not differ significantly (*p* > 0.05) from the natural control. However, in the natural control group, the deposition of pollen on the stigma was minimal, accounting for only 0.04% (7/150), and it could not be confirmed whether the deposited pollen was indeed from *S. nepalense* var. *ciliatum*. No instances of automatic pollen deposition on the stigma were observed in the self-pollination treatment. Therefore, in its natural state, *S. nepalense* var. *ciliatum* primarily relies on apomixis reproductive mechanisms.

In order to further explore whether pollen from different pollination methods germinates, an investigation was conducted on the pollen germination under different bagging methods. The results showed that pollen tube germinated under three different artificial treatments. The results indicated that regardless of whether it was manual self-pollination (Figure 6b), out-crossing pollination (Figure 6c), or geitonogamy (Figure 6d), the pollen in all these treatments germinated, and the pollen tubes entered the ovary.

## 3. Discussion

### 3.1. Pollination Syndrome Characterized by Out-Crossing Pollination including Flowering Period and Floral Features

The flowering period of *Satyrium nepalense* var. *ciliatum* populations in Liangwang Mountain ranges from August to October, with individual flowers lasting approximately 10.75 ± 1.06 days. A long flowering period provides favorable conditions for pollination, while also allowing ample time for plants to adapt to environmental changes, thereby reducing the adverse effects of harsh environmental conditions on pollination and ensuring reproductive success [38,39]. Additionally, the phenology of flowering in Northwestern Yunnan populations of *S. nepalense* var. *ciliatum* has led to floral variation, with short-spurred female individuals exhibiting an adaptive mechanism of earlier flowering and increased fruit set [32]. Flowering phenology is a crucial trait in the reproductive life history of plants, impacting reproductive success and adaptability through pollination and seed dispersal, germination [40,41].

*Satyrium nepalense* var. *ciliatum* has pink flowers reminiscent of its congeners like *S. hallackii*, known to entice bees, dipterans, and moths into the pollinator [22]. This coloration aligns with the pollination syndrome typically associated with butterfly-pollinated species, further broadening the spectrum of potential pollinators [4]. Furthermore, this species has a well-developed rostellum, which plays a crucial role in preventing autonomous self-pollination within the same flower [42,43]. The sides of rostellum bear two plate-like viscidia, facilitating pollen attachment to visiting pollinators [4,42]. Additionally, the reticulate pollen ornamentation of *S. nepalense* var. *ciliatum* enables adhesion to pollinators’ bodies. The pollen exine ornamentation plays a significant role in pollination within Orchidaceae, pollen can adhere to pollinators through the ornamentation [44,45].

Furthermore, the provision of floral rewards, including nectar-releasing stomata, food hairs, and nectaries, serves as a key mechanism to attract and reward flower visitors for *S. nepalense* var. *ciliatum*. Most *Satyrium* species offer nectar, and notably, *S. longicauda* exhibits two pollination systems, with a shift towards oil-bee pollination from moth-nectar pollination [46]. *S. nepalense* var. *ciliatum* is considered to be out-crossing pollination by the flowering phenology and pollination syndromes.

### 3.2. Potential Relationship between Ants and Satyrium nepalense var. ciliatum

*Satyrium nepalense* var. *ciliatum* has nectar and ants enter flowers to feed on the nectar and pollen carriage has also been observed, but whether ants are pollinators remains to be investigated. A few orchid species have indeed been confirmed to be pollinated by ants [47,48,49,50]. Several factors contribute to a low pollination efficiency of ants, including floral mechanisms that deter ants, such as sticky secretions and glandular hairs, along with chemical compounds and extrafloral nectar that prevent ants from entering flowers [51,52,53,54,55]. Ants’ small size, smooth bodies, primarily wingless nature, and self-grooming habits further limit their ability to carry pollen [56,57,58,59,60]. Moreover, substances with antibiotic properties secreted by ant abdominal glands might impact pollen viability and seed fertilization rates, potentially reducing the vitality of both pollen and seeds with proximity to these glands [61,62,63,64]. To conclusively identify ants as pollinators, it is essential to demonstrate their capability to transfer pollen from the anther to their bodies, then from their bodies to the stigma, leading to the subsequent production of fertile seeds [58].

Beyond their putative role in pollination, ants harbor other ecological implications within the *S. nepalense* var. *ciliatum*, and other studies have shown that *Caularthron bilamellatum* (Rchb.f.) R.E.Schult. can obtain various forms of nitrogen from ant nests [65]. In *Dendrophylax lindenii* (Lindl.) Benth. ex Rolfe, ants nest at the plant’s roots, and their excrement provides nutrients to the plant, while the ants act as protectors, defending the plant against potential threats [66]. However, the specific interaction between ants and *S. nepalense* var. *ciliatum* requires further research.

### 3.3. Pollination Syndrome, Florivory, Pollination Competition and Breeding System

Bagging and pollen tube germination experiments have revealed diverse breeding systems in the *Satyrium nepalense* var. *ciliatum* populations in Liangwang Mountain. Field observations and autonomous self-pollination experiments indicate extremely low pollen acceptance rates in nature, with no observed pollen adherence to the stigma automatically. In addition, it exhibits florivory. Hence, in nature, *S. nepalense* var. *ciliatum* primarily adopts apomixis similar to the other populations [32,33].

Additionally, insect pollination behaviors have eluded observation, with only a solitary instance of bee pollination recorded over several years within the Northwestern Yunnan population [33,34], the scarcity of pollinators in its habitat may contribute to its predisposition towards selfing, potentially driven by the scarcity of pollinators in its habitat [33]. It is postulated that the high-altitude distribution of *S. nepalense* var. *ciliatum* relative to other *Satyrium* species contributes to the scarcity of pollinators [33]. However, the habitat under investigation harbors many of the “traditional” pollinators, including bees, moths, and butterflies. Yet, their preference for companion species has resulted in a paucity of pollinators within the *S. nepalense* var. *ciliatum*. Furthermore, the typical characteristic of *Satyrium* flowers pollinated by bees is a spur length of 10 to 15 mm [4], whereas for *S. nepalense* var. *ciliatum*, the spur length is shorter in this site. Therefore, while the species is self-compatible, the developed rostellum makes autonomous self-pollination difficult. Coupled with constrained pollinator availability and florivory, these factors have led to apomixis as the predominant breeding system in nature.

## 4. Materials and Methods

Ethics approval and consent to participate: The study was conducted the plant material that complies with relevant institutional, national, and international guidelines and legislation. The samples collected for this research experiment will not have any impact on the population in nature.

### 4.1. Study Site and Plant Material

Liangwang Mountain (24°43′57″–24°27′28″ N, 102°52′45″–102°55′15″ E), located at the boundary of Southeastern Kunming City and Northeastern Yuxi City in Chengjiang County, Yunnan Province, China, stands as the highest peak in the Central Yunnan, with an elevation ranging from 2000 to 2820 m. The mountainous region falls under the typical temperate climate zone, and its summit, positioned above 2400 m, is frequently veiled in clouds and fog throughout the year, with an average annual temperature of 11.7 °C. This creates a conducive environment for a diverse range of rare and endangered plants [67]. *Satyrium nepalense* var. *ciliatum* is an alpine terrestrial orchid and is found on grassy slopes at elevations ranging from 2256 to 2736 m.

### 4.2. Floral Features

Thirty plants were selected to observe the number of flowers in the inflorescence. The morphological traits of fresh flowers (*n* = 30) of *Satyrium nepalense* var. *ciliatum* were recorded under a Leica M165FC stereomicroscope (Leica, Wetzlar, Germany), and the length and width of their bracts, lips, petals, median sepals, lateral sepals, and flower channel were measured by ImageJ [68,69], and the length of the flower spur was measured using a vernier caliper with a precision of 0.05 mm. Nectar volume was measured using 2 μL Minicaps (Hirschmann, Eberstadt, Germany) at five time points (9:00 a.m., 11:00 a.m., 1:00 p.m., 3:00 p.m., and 5:00 p.m.). Flowers (*n* = 100) from three different plants were sampled at each time point, with nectar volume calculated as the sum of nectar collected from both sides of the floral spur. Statistical analyses were performed using Excel 2016 and means and standard deviations were calculated (mean ± SD), retaining two decimal places.

Flowers at the anthesis stage were fixed in formalin, acetic acid, and alcohol (FAA, 50% alcohol/acetic acid/formaldehyde = 90:5:5) for anatomical and scanning electron microscopy (SEM) examination. Alcohol, acetic acid and formaldehyde were manufactured by the Tianjin Zhiyuan Chemical Reagent Co., Ltd. (Tianjin, China). 

For preparing permanent microscope sections, the fixed material was processed by dehydration through an ethanol series with a pre-impregnant rinsing of xylenes (Tianjin Zhiyuan Chemical Reagent Co., Ltd., Tianjin, China), and infiltration in paraffin. Floral pieces were sectioned transversely (10 μm thickness) with a Canada balsam (Beijing Labgic Technology Co., Ltd., Beijing, China); the sections were stained with hematoxylin (Coolaber Co., Ltd., Beijing China). Some material was subjected to maceration using a solution of H_2_O_2_-CH_2_COOH (Xilong Scientific Co., Ltd., Guangzhou, China) at 60 °C for 24~48 h, then hand-cut sections were made and stained with the Periodic Acid-Schiff (PAS), and Fehling’s solution (BiobomeiBiotechnology Co., Ltd., Anhui, China). The sections were then observed and photographed using a Nikon E100 light microscope (Nikon, Tokyo, Japan).

Floral pieces were first dehydrated through a series of increasing ethanol solutions. Then, they were critical point dried with solvent substituted liquid carbon dioxide and coated with a thin layer of gold–palladium. Finally, the samples were examined with a Scanning Electron Microscope EVO LS10 (Zeiss, Oberkochen, Germany).

### 4.3. Florivory and Flower Visitor Observation

In 2022 and 2023, the florivory of *Satyrium nepalense* var. *ciliatum* was observed (50 plants). The flower visitors were recorded by a Nikon D3500 camera (Nikon, Tokyo, Japan) lasting 100 h, from 8:00 to 18:00, subdivided into hourly intervals. Additionally, selected inflorescences were dissected for closer examination to determine the presence of inconspicuous flower visitors. The flower visitors, visiting companion species, were also recorded. To determine if the ants carried pollinia, 10 ants of each species were captured and observed using a Leica M165FC stereomicroscope (Leica, Wetzlar, Germany). The mean ± SD of the different flower visitors at each time period during the observation time were calculated using Excel 2016 and statistical plots were made using Origin 2018.

### 4.4. Breeding System

This study implemented seven artificial pollination treatments on *Satyrium nepalense* var. *ciliatum*: (1) autonomous self-pollination (*n* = 53), wherein flowers were directly bagged; (2) manual self-pollination (*n* = 48), involving the manual transfer of pollen from a flower’s anther to its own stigma; (3) manual geitonogamy (*n* = 58), entailing artificial cross-pollination among flowers from the same inflorescence; (4) manual out-crossing (*n* = 39), consisting of the manual transfer of pollen between flowers of different inflorescences; (5) apomixis (*n* = 70), involving the removal of both stigma and anther followed by bagging to prevent any form of sexual reproduction; and (6) natural control (*n* = 150), characterized by the absence of any treatment. Chi-square tests were employed for the statistical analysis of fruit set rates. Fruit set was calculated using Excel 2016 and significant differences between different artificial pollination methods and the natural state were calculated using Chi-square tests.

### 4.5. Pollen Tube Germination Experiment

After manual pollination (self-pollination, out-crossing, and geitonogamy) at 8, 12, 24, 48, 72, and 96 h, flowers were fixed in Carnoy’s fixative (Anhydrous ethanol/glacial acetic acid = 3:1) and underwent rehydration in 70%, 50%, and 30% ethanol, followed by overnight immersion in distilled water at 4 °C. Pollen tube growth was observed using the improved aniline blue staining method [70]. Flowers were cleared in 8 N NaOH (Guanghua Sci-Tech Co., Ltd., Shantou, China) overnight at room temperature, washed three times with distilled water, and left in distilled water overnight. Finally, the flowers were stained with 0.1% Aniline blue in 0.1 M K_3_PO_4_ buffer (Macklin Biochemical Technology Co., Ltd., Shanghai, China) and 2% glycerol (*v/v*) (Guanghua Sci-Tech Co., Ltd., Shantou, China) in darkness overnight.

## 5. Conclusions

In summary, *Satyrium nepalense* var. *ciliatum* in Liangwang Mountain exhibits an out-crossing preference in pollination, as indicated by its flowering phenology and floral traits. Despite adaptations favoring out-crossing, such as a long flowering period and floral rewards, limited observed insect pollination suggests a reliance on self-pollination possibly, especially in environments with few pollinators. Factors like low pollen acceptance rates and florivory further support apomixis as the primary reproductive strategies. Thus, the species shows selfing and out-crossing, while pollinator scarcity and florivory lead to apomixis. Further investigation into pollinator interactions and their ecological implications can enhance the understanding of *Satyrium* adaptation in challenging environments.

## Figures and Tables

**Figure 1 plants-13-01228-f001:**
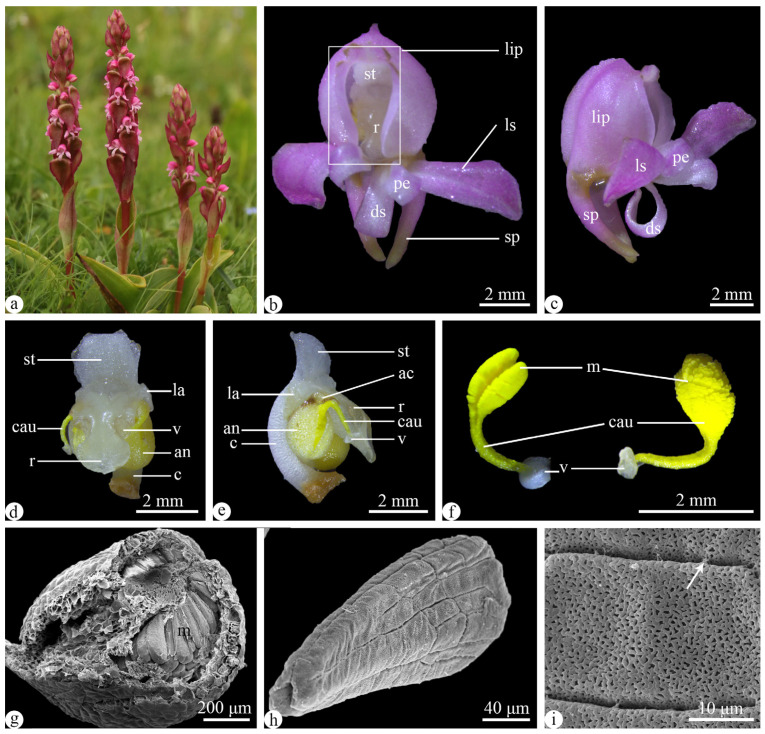
Floral morphology of *Satyrium nepalense* var. *ciliatum*. (**a**) Plants. (**b**,**c**) Front (**b**), and side (**c**) views of a flower, noting the flower channel (box in (**b**)). (**d**,**e**) Front (**d**), and side (**e**) views of a gynostemium. (**f**) A pair of sectile pollinia with yellow caudilce and sticky viscidium. (**g**) Cross-section of an anther under the SEM, showing the arrangement of massulae. (**h**,**i**) A massula (**h**), and exine sculpture (**i**) under the SEM, noting viscous substance (arrow). Abbreviations: ac = anther channel, an = anther, c = column, cau = caudicle, ds = dorsal sepal, la = lateral appendages, ls = lateral sepal, m = massulae, pe = petal, r = rostellum, st = stigma, sp = spur, v = viscidium.

**Figure 2 plants-13-01228-f002:**
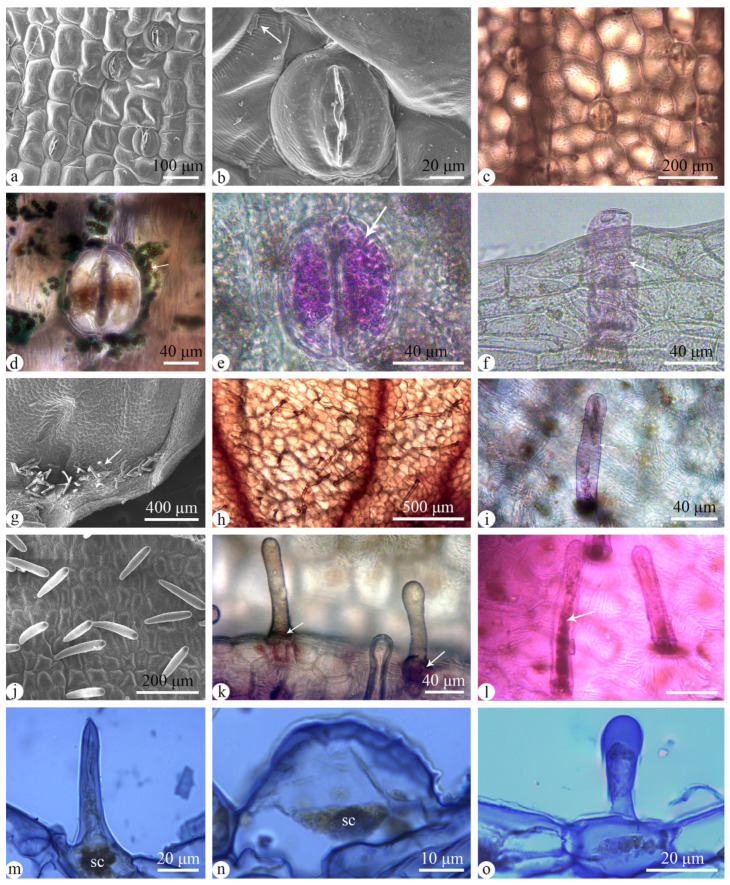
Micromorphology of the bract and the inner wall of the spur of *Satyrium nepalense* var. *ciliatum*. (**a**–**f**) Adaxial surface of the bract under the SEM (**a**,**b**), and under the light microscope stained with Fehling’s solution (**c**,**d**), and PAS (**e**,**f**), showing sugar (arrows). (**g**–**i**) Food hairs (arrow) at the lip base under the SEM (**g**) and under the light microscope, stained with Fehling’s solution (**h**), and PAS (**i**), sugar (arrow). (**j**–**l**) The inner wall of the spur under the SEM, showing nectar glands (**j**), and under the light microscope stained with Fehling’s solution (**k**), and PAS (**l**), sugar (arrows). (**m**–**o**) Three types of nectar glands under the microscope, including villiform (**m**), dome-shaped (**n**), and obpyriform (**o**). Abbreviations: sc = secretory cell.

**Figure 3 plants-13-01228-f003:**
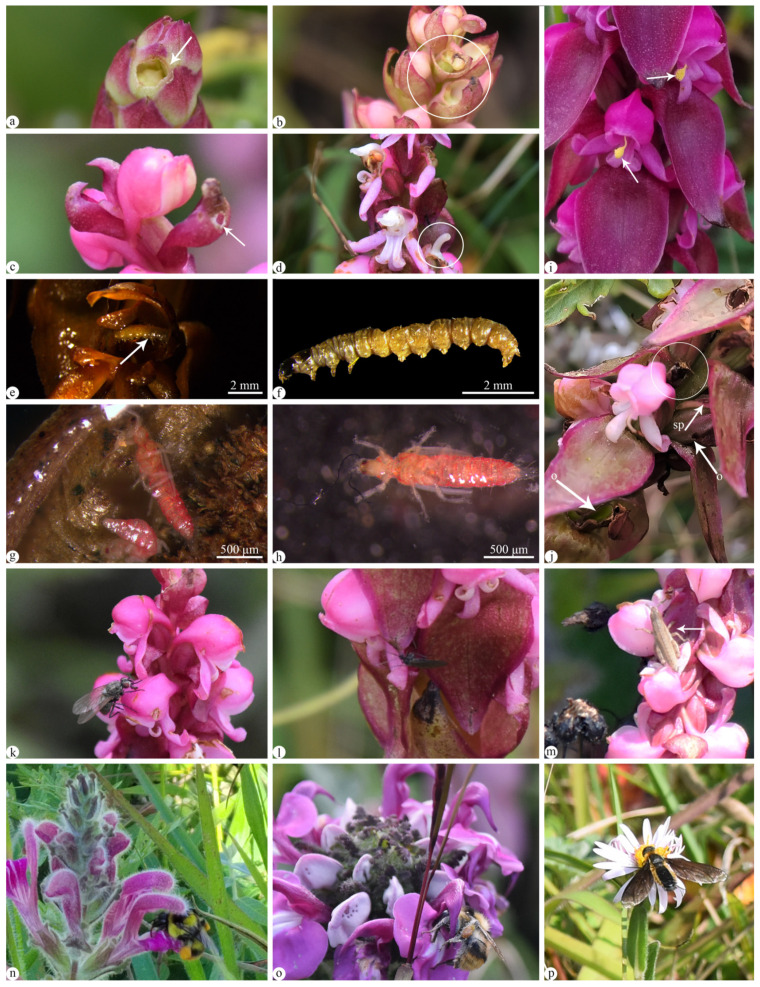
Florivory and some flower visitors of *Satyrium nepalense* var. *ciliatum*. (**a**–**d**) Florivory at developing stages of flowers, noting florivory (arrows and circles) occurred on lip, bract, and stigma. (**e**,**f**) A worm (arrow) found in a florivorous flower. (**g**,**h**) Thripidae dissected in detected ovaries. (**i**) Two sectile pollinia observed outside the flower and adhered to the lip and petal (arrow). (**j**) A beetle (circle) observed with florivory on the ovary and spur. (**k**–**m**) Some flower visitors, a fly (**k**), Diptera (**l**), a locust larva (**m**). (**n**–**p**) Some insects visiting the companion species, bumblebee visiting *Pedicularis salviiflora* (**n**), bumblebee visiting *Pedicularis gruina* (**o**), a Diptera visiting *Aster yunnanensis* (**p**). Abbreviations: sp = spur, o = ovary.

**Figure 4 plants-13-01228-f004:**
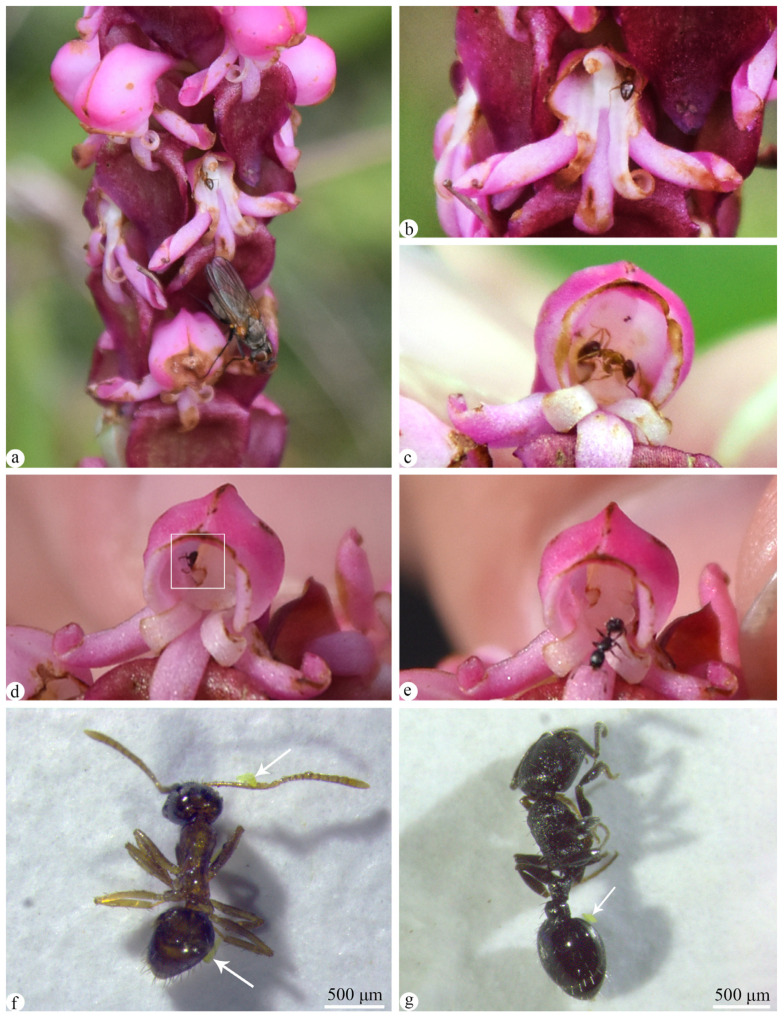
Two species of ants visiting flowers of *Satyrium nepalense* var. *ciliatum*. (**a**) *Pheidole zoceana* entered the left spur with a fly on a florivorous flower. (**b**) *P. zoceana* entered the right spur of a florivorous flower. (**c**). *P. zoceana* passed through the stigma in a flower. (**d**) *Temnothorax* sp. entered the spur of a flower. (**e**) *Temnothorax* sp. out of a flower. (**f**) Massulae attached to the body of *P. zoceana*. (**g**) Massulae attached to the body of *Temnothorax* sp.

**Figure 5 plants-13-01228-f005:**
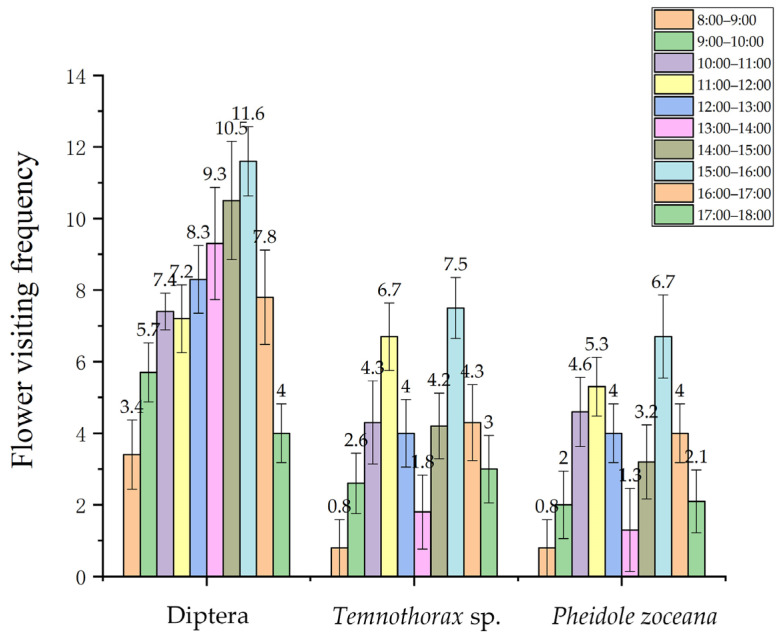
Flower visiting frequency of three flower visitors (mean ± SD).

**Figure 6 plants-13-01228-f006:**
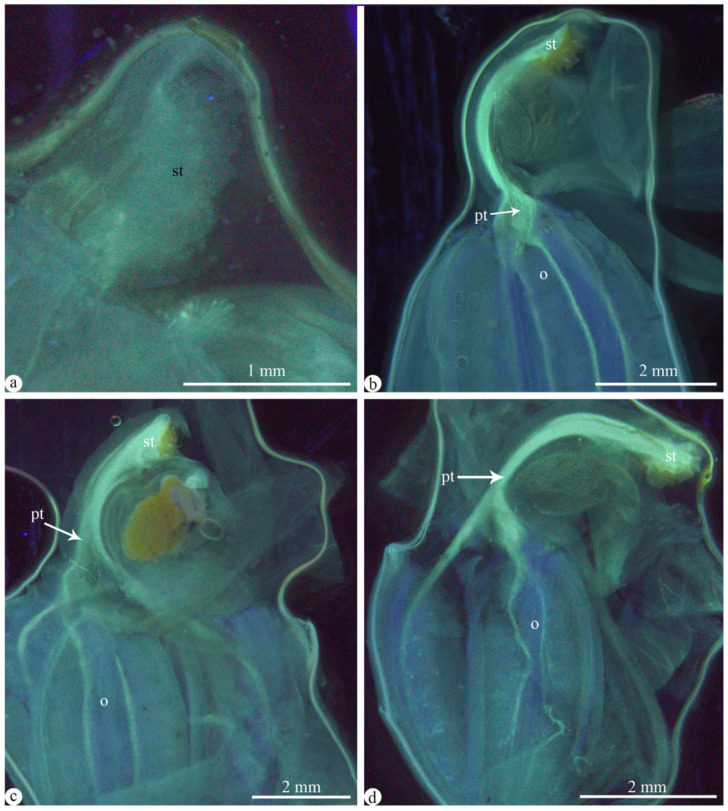
Pollen germination and growth on the stigma and through the gynostemium to the ovary under four different artificial pollination treatments. (**a**) Un-pollination. (**b**) Self-pollination. (**c**) Out-crossing pollination. (**d**) Geitonogamy. Abbreviations: o = ovary, st = stigma, pt = pollen tube.

## Data Availability

The data will be made available by the authors on request.

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
