# Peer review of "Pollination Syndrome, Florivory, and Breeding System of Satyrium nepalense var. ciliatum (Orchidaceae) in Central Yunnan, China"

_plants, 2024, doi:10.3390/plants13091228_

Round 1

Reviewer 1 Report

Comments and Suggestions for Authors

Dear Authors,

I read with great interest your paper sent to Plants with the title: Pollination syndrome, florivory, and breeding system of Satyrium nepalense var. ciliatum (Orchidaceae) in Center Yunnan, China.

The genus Satyrium is known for its unsurpassed floral diversity, particularly for pollinator-mediated pollination systems. Consistent with the floral traits, most members of the genus exhibit a specialized biotic pollination system, including bees, beetles, birds, butterflies, flies, and moths as specialized pollinators. S. nepalense, like other members of the genus, possesses floral traits consistent with a pollinator-mediated pollination system. The pollination biology of this species is fairly well known. However, not much data has been published on the pollination biology of intraspecific taxa. Therefore, I encourage the authors to explain in detail in the introductory chapter what characteristics distinguish the S. nepalense var. ciliatum. In the discussion section, it should be clearly stated whether the S. nepalense var. ciliatum differs from other varietas or subspecies of Satyrium nepalense in terms of pollination biology. Are the data presented new to the science, or do the investigations of the authors confirm facts that are generally known?

I also have serious doubts about the data presented in Figure 5. I know from experience that accurate determination of nectar volume is very difficult; it is dependent on the condition of the individuals under study, the ambient temperature, and many other factors. Typically, we study a very large number of individuals and average the results. How exactly the authors determined the nectar volume and how many individuals were examined - these data are not included in the Materials and Methods chapter. Please complete the methodology. Does the nectar volume of the examined variety differ from data for other intraspecific taxa?

I also have a few minor comments:

Line 54: Please replace “(references in Johnson et al. 2011). “ with "(references in [x])". X is the number of cited work in references

Lines 72, 107, 348, and 374: Latin names of taxa are always written in italics

Line 283: in header 3.2 the abbreviation var. should be written without italics - I don't know if this can be changed due to MDPI publishing recommendations, but formally it is an error.

Comments on the Quality of English Language

The English needs to be polished by a native English speaker.

Reviewer 2 Report

Comments and Suggestions for Authors

The article under review deals with important questions on pollination of Orchidaceae flowers and presents the results of an interesting study. Without going into the strengths of the article, I would like to make some comments on what should be corrected in the article. 

First of all, I would like to express my doubts about the use of the term 'syndrome'. Despite the fact that it is sometimes used in some publications and has become fashionable, it is misleading. The use of the term 'pollination syndrome' cannot be justified either semantically or in terms of usage. The basic meaning of the term 'syndrome' is 'signs and symptoms that occur together and characterise a particular abnormality or condition'. There is no abnormality in the analysed phenomenon. The use of a term with strong medical or pathological connotations is, in my view, unjustified. Instead, I suggest looking for a term that better describes the event logically and semantically, such as 'pattern'. On the other hand, I think that instead of the term 'characteristics', it is better to use the term 'traits', which more clearly defines the analysed flower structures and their properties.

1. The authors of the name of the taxon in consideration should be indicated in the abstract (line 15) and in the text. It is also necessary to indicate the authors of the names of other taxa (at their first mentioning) to ensure an unambiguous understanding of the taxa the authors are writing about. 

2. The objectives of the study state that the authors aim to "analyse the pollination syndromes of Satyrium humile" (line 107), which is native to Cape Province. So which taxon were the authors actually studying?

3. In this article, there is absolutely no justification for moving the methodology to the end of the article. Without reading the methodology and material, the results are incomprehensible. It is necessary to move the methodology after the introduction. Moreover, in the journal Plants, many articles are published with a logical structure, with the methodology preceding the results.

4. How to understand the statement 'approximately 22 flowers'? Logically there cannot be 'approximately 22 flowers'. Are you writing about the mean number of flowers? If so, this must be precise and must be described in the methodology.

5. To what precision were the measurements made? The question arises because the means are given with two decimal places. This must be described in the methodology in order to avoid further questions. The descriptions of the statistical methods are limited to a single mention of the chi-squared test. What about other calculations and comparisons? I believe that these methods of analysis are not sufficient to draw valid conclusions. 

6. In my opinion, there are too many illustrations in the results section. I do not deny that they are of good quality and useful, but they are not all necessary. Moreover, some of the phenomena depicted in the photographs are only very briefly mentioned in the results section.

7. Can 'syndrome' really attract insects (line 394)? I have the impression that some parts of the text are written in jargon that only the authors understand. Scientific articles must be written in clearly worded sentences that express the exact idea and describe the phenomenon.

The article has great potential, but needs to be restructured, especially the methodology and results sections. Moreover, the results section gives the impression that the different parts of the study are not interconnected, although there are clear links and logic between them which need to be emphasised when the article is reorganised.

Comments on the Quality of English Language

Moderate editing of the English language is required, with a particular emphasis on tenses. There are quite a lot of technical shortcomings (out-of-place full stops, missing italics, etc.). The most serious flaw is that some sentences are linguistically correct but logically imprecise. 

Reviewer 3 Report

Comments and Suggestions for Authors

Dear Authors,

The manuscript investigates the characteristics of the flowers of Satyrium nepalense var. ciliatum, an alpine orchid, along with its pollen germination, flower visitors, and pollination mechanism in Liangwang Mountain, China. The study provides a detailed description of flower structure and morphology, analyzes flower visitors, and examines the phenomenon of pollination. Based on the research results, it is concluded that the orchid exhibits a cross-pollination flower syndrome, and ants were observed among the flower visitors for the first time, although further research is needed to determine their exact role. Additionally, the abstract highlights that the pollination syndrome, florivory, and pollination competition contribute to the plant's mixed reproductive system, particularly promoting apomixis.

Abstract: The introduction should be more concise to provide a better overview of the main results and conclusions of the study.

Introduction: While the introduction includes the study's purpose and research background, more information is needed on the context of the work and the significance of the plant species under study.

Results: The results are detailed and thorough, but the text can be difficult to read at times and may be overly detailed in some sections.

Discussion: The discussions are comprehensive and well-developed; however, more emphasis should be placed on establishing connections and the practical significance of the research findings.

Materials and Methods: The materials and methods seem inadequately described, and some technical details may be difficult for readers to understand. Unfortunately, no statistical information is mentioned in the materials and methods section, nor in the relevant figures. For example: What statistics were used in the study? What was the sample number? What does ± mean, SE or SD?

Conclusion: The conclusion summarizes the main findings and the significance of the study, but it mostly repeats the results and discussion. It requires a more detailed summary of the main conclusions and possible directions for future research.

Since it is a protected species, the authors must make a statement about the influence the sampling had on the studied population.

Minor errors:

Line 47, 51, 72, 107, 330, 374: Satyrium shoul be writen in italics. Please check it throught the MS.

L 55-56: please name the family of sunbirds in parentheses (Nectarinidae).

L 176: ‘Figure 6k’ instead of ‘Figure 6: k’. Please check it throught the MS.

L 192: What do you mean by (Figure 6: n ~ p)??

L 289: Double space.

L 204-205, 346-347, 366-367, 372-373, 383-384: An empty line is missing.

L 335: an ‘m’ is missing. ‘ciliatum’ instead of ‘ciliatu’

L 350: A space is missing.

L 360: The font style is wrong ..°C..

Fig 5: It would be worthwhile to use a different color for the diagrams in the figure. Why were statistics not used? The figure legend should be supplemented: What was the sample number? Are the data in mean and SD or SE?

Fig 8: Why were statistics not used? The figure legend should be supplemented: What was the sample number? Are the data in mean and SD or SE? The names of genera and species should be in italics on the x-axis of the figure. The captions on both axes can be in larger font size.

While some methodological details and text editing need improvement, the research provides a fundamentally sound scientific basis and valuable results.

Comments on the Quality of English Language

The quality of English grammar is generally appropriate for scientific communication, but sentence structure may sometimes be complex, and some details may be difficult for readers to understand. Improvements could include simplifying sentences, making the text more accessible, and better structuring details and emphasizing connections.

Round 2

Reviewer 1 Report

Comments and Suggestions for Authors

Compared to the original version, the authors have significantly improved the manuscript. This is a much better version of the manuscript than the previous submission. I have no further comments. My recommendation is for the publication of this article.

Reviewer 2 Report

Comments and Suggestions for Authors

The revised and expanded version of the article is significantly improved. Most of the comments made in the review have been addressed or clarified by the authors. Nevertheless, there are still points to be corrected and new inaccuracies have arisen in the revision process.

I understand the authors' reluctance to replace the semantically inaccurate term 'pollination syndrome' with a semantically and conceptually more accurate term, but I do not agree with their decision. Nevertheless, it is up to the authors of the article to make their decision and I must respect it. I would just like to add that one of the most important principles of terminology is that any term cannot semantically contradict the derivational basis of a term. In this case, the term 'syndrome' has no relation, either etymologically or in terms of its meaning in usage, to the phenomenon being defined. 

How many and which 'pollination syndromes' did the authors study? The abstract (line 16, line 49, line 112, etc.) refers to 'pollination syndromes' (plural), which raises the question of whether 'pollination syndrome' is understood by the authors as a whole, or whether it is divided into variants of a syndrome? 

I must add that the use of the plural in many places in the text of the article raises serious doubts. For example (line 85) it says: 'Pollination syndrome is linked to breeding systems'. This poses the question: where are the 'breeding systems'? In my opinion there is a breeding system with several variants (selfing, outcrossing, apomixis). Do the authors think that each variant is a separate system? In that case, another question arises: how do the authors understand the term 'system'?  I think the authors simply pay too little attention to the logical meaning and significance of linguistic resources. 

What do the authors mean by 'Insects were attracted to the plant [...] (line 21). What attracted the insects to the plant? I think it is necessary to consider each statement, especially those in the abstract. 

It is very difficult to understand the logic of the authors when the text interweaves various unrelated things. For example, lines 38-41 describe the differences between the varieties of the species studied, and then (from line 41 onwards) begin to provide information on the ethnopharmacological use of the plant. If this information is considered important, it should be properly separated, perhaps in a separate paragraph. In my opinion, the morphological differences between the subspecies are completely irrelevant to their pharmacological properties. 

Perhaps the authors will accuse me of harassment, but the statement that 'is scentless to the human nose' (line 140) is a vernacular expression, not a scientific one. If we want to be precise, we should say that the human sense of smell does not detect the smell of flowers, or that humans cannot detect the smell of flowers. 

In my previous review I wrote that some of the illustrations are not linked to the text (even though they are really good). In this version, there are the rest of the illustrations, and the references are confusing. For example, 'In addition, pollen was naturally exposed and adhered to the lip and petals (Figure 3: i, j)' (line 174-175), but the picture does not show how the pollen 'adhered to the lip and petals'. In one picture (i), 'exposed' pollinaria can be seen. 

I very much miss the statistical interpretation of the data. For example, the graph in Figure 5 should indicate between which time periods there are significant differences and between which there are no differences. A more in-depth analysis could be carried out using permutation analysis to assess the determinants of the intensity of pollinator visits.

Comments on the Quality of English Language

The English of the article is correct, with only minor corrections needed, but the use of subjects and plurals is flawed, leading in some cases to factual inaccuracies or even logical errors. 

Reviewer 3 Report

Comments and Suggestions for Authors

I find the corrected manuscript adequate.
